METHODS AND RESOURCES

# A new test suggests hundreds of amino acid polymorphisms in humans are subject to balancing selection

**Vivak Soni** [1⊕], **Michiel Vos** [2⊕], **Adam Eyre-Walker** [1] *

**1** School of Life Sciences, University of Sussex, Brighton, United Kingdom, **2** European Centre for Environment and Human Health, University of Exeter Medical School, Environment and Sustainability Institute, Penryn, United Kingdom

⊕ These authors contributed equally to this work.
* a.c.eyre-walker@sussex.ac.uk

## Abstract

The role that balancing selection plays in the maintenance of genetic diversity remains unresolved. Here, we introduce a new test, based on the McDonald–Kreitman test, in which the number of polymorphisms that are shared between populations is contrasted to those that are private at selected and neutral sites. We show that this simple test is robust to a variety of demographic changes, and that it can also give a direct estimate of the number of shared polymorphisms that are directly maintained by balancing selection. We apply our method to population genomic data from humans and provide some evidence that hundreds of nonsynonymous polymorphisms are subject to balancing selection.

## Introduction

How genetic variation is maintained, either in the form of DNA sequence diversity or quantitative genetic variation, remains one of the central problems of population genetics. Balancing selection encapsulates several selective mechanisms that increase variability within a population. These include heterozygote advantage (also referred to as overdominance), frequency-dependent selection, and selection that varies through space and time [1]. However, although there are some clear examples of each type of selection [2,3], the overall role that balancing selection plays in maintaining genetic variation, either directly or indirectly through linkage, remains unknown.

Numerous methods have been developed to detect the signature of balancing selection [4–15]. Application of these methods have identified a number of loci subject to balancing selection, largely in the human genome, in which most of this research has taken place. However, many of these methods are quite complex to apply, often leveraging multiple population genetic signatures of balancing selection and requiring simulations to determine the null distribution. Furthermore, they do not readily yield an estimate of the number of polymorphisms that are directly subject to balancing selection, as opposed to being in linkage disequilibrium

**Data Availability Statement:** We have used the publicly available 1000 genome data available at https://www.internationalgenome.org.

**Funding:** This research was supported by National Environment Research Council (NERC) grant NE/T008083/1 to author MV. URL: https://nerc.ukri.org/funding/next/publicationofwork/ The funders had no role in study design, data collection and analysis, decision to publish, or preparation of the manuscript.

**Competing interests:** The authors have declared that no competing interests exist.

**Abbreviations:** BGC, biased gene conversion; DFE, distributions of fitness effect; GO, gene ontology; HLA, human leukocyte antigen; LD, linkage disequilibrium; MAF, minor allele frequency; MHC, major histocompatibility complex; RR, recombination rate; SDMs, slightly deleterious mutations; SFS, site frequency spectrum; tMRCA, time to most common recent ancestor.

(LD) with them. Here, we introduce a method that is simple to apply and which generates a direct estimate of the number of polymorphisms subject to balancing selection.

One signature of balancing selection that has been utilised in several studies is the sharing of polymorphisms between species [5,8,10]. If the species are sufficiently divergent that they are unlikely to share neutral polymorphisms, then shared genetic variation can be attributed to balancing selection. These studies have concluded that there are relatively few balanced polymorphisms that are shared between humans and chimpanzees [5,8]. However, this test is likely to be weak because humans and chimpanzees diverged millions of years in the past, and it is unlikely that any shared selection pressures will be maintained over that time period.

The major problem with approaches that consider the sharing of polymorphisms between species or populations is differentiating selectively maintained polymorphisms from neutral variation inherited from the common ancestor. This problem can be solved by comparing the number of shared polymorphisms at sites that are selected, to those that are neutral. We expect the number of shared polymorphisms at selected sites to be lower than at neutral sites because many mutations at selected sites are likely to be deleterious, and hence unlikely to be shared. However, we can estimate the proportion that are effectively neutral by considering the ratio of polymorphisms, which are private to one of the 2 populations or species, at selected versus neutral sites. Although the method can be applied to any group of neutral and selected sites that are interspersed with one another, we will characterise it in terms of nonsynonymous and synonymous sites. Let the numbers of polymorphisms that are shared between 2 populations or species be $S_N$ and $S_S$ at nonsynonymous and synonymous sites, respectively, and the numbers that are private to one of the populations be $R_N$ and $R_S$, respectively. Let us assume that synonymous mutations are neutral and nonsynonymous mutations are either neutral or strongly deleterious. Then, it is evident that $\frac{S_N}{S_S} = \frac{R_N}{R_S} = f$, where $f$ is the proportion of the nonsynonymous mutations that are neutral. However, if there is balancing selection acting on some nonsynonymous SNPs, and this selection persists for some time such that the balanced polymorphisms are shared between populations then $\frac{S_N}{S_S} > \frac{R_N}{R_S}$. A simple test of balancing selection is therefore whether $Z > 1$, where

$$Z = \frac{S_N/S_S}{R_N/R_S},$$ 
(1)

a simple corollary of the McDonald–Kreitman test for adaptive divergence between species [16]. It can be shown, under some simplifying assumptions in which synonymous mutations are neutral and nonsynonymous mutations are strongly deleterious, neutral or subject to balancing selection, that an estimate of the proportion of nonsynonymous mutations subject directly to balancing selection is $\alpha_b = 1 - \frac{S_S R_N}{S_N R_S}$ (see Results section). In this analysis, we perform population genetic simulations to investigate whether the method can detect the signature of balancing selection and assess whether the method is robust to demographic change. Second, we apply the method to human population genetic data. We estimate that substantial numbers of nonsynonymous polymorphisms are likely being maintained by balancing selection in humans.

## Results

### Simulations

We propose a new test for balancing selection in which the ratio of selected to neutral polymorphisms is compared between those that are shared between populations or species and those that are private to populations or species. To explore the properties of our method to

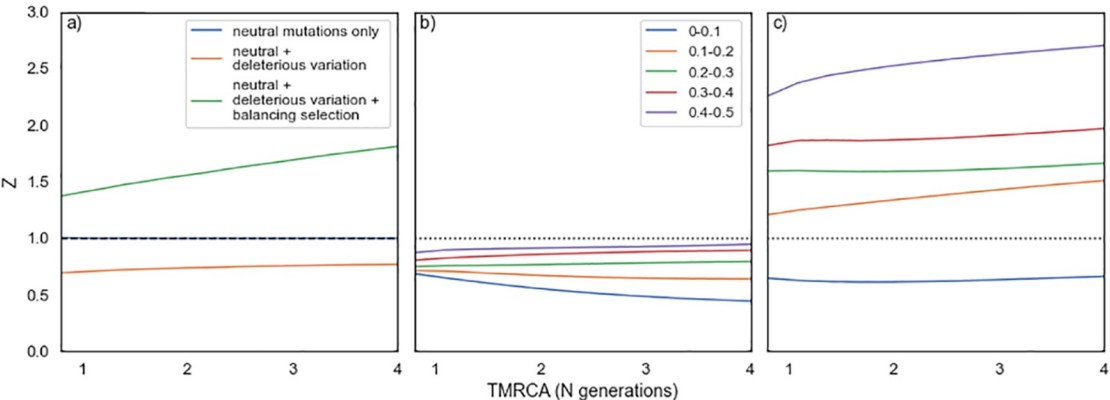

**Fig 1. Stationary population size simulations.** The ancestral population is duplicated to form 2 daughter populations of the same size to each other and the ancestor. The tMRCA is measured in N generations, where N is the population size. In panel (a), we show the value of Z as a function of the tMRCA for 3 scenarios: all nonsynonymous mutations are neutral; all nonsynonymous mutations are deleterious; and all nonsynonymous mutations are neutral except for a single balanced polymorphism in the middle of the locus. In panels (b) and (c) polymorphisms have been binned by minor allele frequency, in bins of size 0.1. In panel (b), we show the case where all nonsynonymous mutations are deleterious and panel (c) all nonsynonymous mutations are deleterious except for a single balanced polymorphism in the middle of the locus. Code to perform these simulations can be at https://github.com/vivaksoni/test_for_balancing_selection. tMRCA, time to the most recent common ancestor.

detect balancing selection, we ran a series of simulations in which an ancestral population splits to yield 2 descendent populations. We initially simulated loci under a simple stationary population size model where the ancestral population is duplicated to form 2 equally sized populations (equal to each other and the ancestral population). This is an unrealistic scenario, but it has the advantage that it involves no demographic change in the transition from ancestral to descendent populations. We assume that synonymous mutations are neutral, and we explore the consequences of different selective models for nonsynonymous mutations. If all nonsynonymous mutations are neutral, then as expected $Z = 1$ (Fig 1a), and if we make some of the nonsynonymous mutations deleterious, drawing their selection coefficients from a gamma distribution, as estimated from human polymorphism data [17] we find that $Z < 1$ (Fig 1a). Again, this is expected because slightly deleterious mutations (SDMs) are likely to contribute more to the level of private than shared polymorphism. If we simulate a locus in which most nonsynonymous mutations are deleterious, drawn from a gamma distribution, but each locus contains a single balanced polymorphism that is shared between populations, then $Z > 1$ (Fig 1a). It is important to note that the density of balanced polymorphisms (i.e., the number per bp) is substantial in these simulations because we have simulated a short exon, of just 288 bp, the average length in humans [18], and each one contains a balanced polymorphism. If we were to reduce the density of balanced polymorphisms, then Z could be less than 1 even if there is balancing selection operating.

SDMs tend to depress the value of Z because they are more likely to segregate within a population than to be shared between populations that diverged sometime in the past; this will tend to make our test (i.e., whether Z > 1) conservative. There are 2 potential strategies for coping with this tendency. We can test for the presence of balancing selection as a function of the frequencies of the polymorphisms in the population, because SDMs will tend to be enriched among the rarer polymorphisms in the population. A similar approach has been used successfully to ameliorate the effects of SDMs in the classic MK approach for estimating the rate of adaptive evolution between species [19–21]. Or we can explicitly model the generation of shared and private polymorphisms under a realistic demographic and selection model to

control for the effects of SDMs. We focus our attention here on the first of these strategies, although we touch on the latter strategy in the discussion. We apply the frequency filter to both the private and shared polymorphisms; this is necessary because if we applied the filter only to the private polymorphisms, we could be comparing high frequency private polymorphisms, with a low ratio of $R_N$ to $R_S$, because SDMs have been excluded, to low frequency shared polymorphisms, which may contain many SDMs and hence have a high value of $S_N/S_S$; this can yield artefactual evidence of balancing selection. This could be exacerbated if some of the SDMs are recessive. For shared polymorphisms, we estimated their frequency in the population from which the private polymorphisms are drawn. To investigate the effects of polymorphism frequency on our estimate of Z, we divided polymorphisms into 5 bins of 0.1 (we did not orient SNPs). If we simulate a population in which nonsynonymous mutations are deleterious, whose effects are drawn from a gamma distribution, we find that $Z < 1$ but this is less marked for the high frequency categories, as we expect (Fig 1b). For the lowest frequency category, Z decreases as a function of the time to most recent common ancestor, whereas for the higher frequency categories, it is either unaffected or increases slightly (Fig 1b). If we include a balanced polymorphism, introduced prior to the population split and subject to strong selection, into the model, which still also includes deleterious mutations, we find that $Z > 1$ for all frequency bins except the lowest one (Fig 1c). Note, once again that the level of balancing selection in these simulations is substantial because every locus contains a balanced polymorphism.

The simulation above does not take into account the demographic effects that a division in a population involves. We therefore performed more realistic simulations that involve vicariance and dispersal scenarios with and without migration between the sampled populations (S1–S13 Figs). We also simulated with and without expansion after separation. We performed all simulations under 2 distributions of fitness effects (DFEs), which were estimated from human and *Drosophila melanogaster* populations. In the vicariance scenario, the ancestral population splits into 2 daughter populations of equal or unequal sizes. In the dispersal scenario, a single daughter population is generated by duplicating part of the ancestral population, which remains the same size as it was before; we vary the daughter population size. In both cases, we explore the consequences of expansion after separation of the populations, and we explore the consequences of migration between the 2 populations.

None of the simulated demographic scenarios is capable of generating Z values greater than 1 under either DFE—i.e., the method does not seem to generate false positives (S1–S13 Figs). However, it is worth noting that a more severe difference in the size of the descendant populations results in depressed Z values in the smaller of the 2 populations, demonstrating that demography can affect the value of Z. In all cases, the value of Z is smallest for the lowest frequency category, those polymorphisms with frequencies <0.1, and this frequency category often shows a dramatic difference to the other categories. We therefore suggest combining the polymorphisms above 0.1 when data are limited. As expected, we find that $Z < 1$ in all simulations when we sum all polymorphisms with frequencies >0.1 (S14 and S15 Figs).

## Statistical tests

We can test for balancing selection by testing whether Z is significantly greater than 1, since Z is expected to be 1 when all mutations are neutral, and less than 1 when some nonsynonymous mutations are slightly deleterious. To test for statistical significance at the single gene level, we recommend using a simple chi-squared test of independence on the $2 \times 2$ contingency table that is formed from $S_N$, $S_S$, $R_N$, and $R_S$; this is appropriate given that nonsynonymous and synonymous sites share the same genealogies. For analyses involving more than 1 gene, we

recommend summing the values $S_N$, $S_S$, $R_N$, and $R_S$ across genes and bootstrapping at a level that encompasses all sources of possible variance to derive confidence intervals. In many species, this will be at the gene level. For example, in humans, gene density is such that there is little linkage between genes—there is approximately 1 gene every 150 kb and the average half-life of LD is approximately 20 kb [22].

### Estimating the level of balancing selection

One of the great advantages of our method is that it gives an estimate of the number of polymorphisms that are directly affected by balancing selection under a simple model of evolution. Let us assume that synonymous mutations are neutral and that nonsynonymous mutations are strongly deleterious, neutral, or subject to balancing selection; we further assume that all balanced polymorphisms arose before the 2 populations split. Then, the expected numbers of nonsynonymous, $R_N$, and synonymous, $R_S$, private polymorphisms are

$$
\begin{aligned}
R_s &= \theta \rho W \\
R_N &= \theta \rho W f,
\end{aligned}
\tag{2}
$$

where $\theta = 4 N_e u$, $N_e$ is the effective population size, and $u$ is the mutation rate per site per generation. $\rho$ is the proportion of polymorphisms that are private to the population, $W$ is Watterson's coefficient, and $f$ is the proportion of nonsynonymous mutations that are neutral, $(1-f)$ being deleterious or subject to balancing selection.

In deriving expressions for $S_N$ and $S_S$, we have to take into account that a balanced polymorphism can maintain neutral variation in LD that may also be shared between populations. If we have $b$ balanced nonsynonymous polymorphisms and each of those maintains $x$ neutral mutations in LD, then the expected values of $S_N$ and $S_S$ are

$$
\begin{aligned}
S_S &= \theta(1-\rho)W + bx \\
S_N &= \theta(1-\rho)Wf + b + bxf.
\end{aligned}
\tag{3}
$$

It is then straightforward to show that the proportion of shared nonsynonymous polymorphisms that are directly maintained by balancing selection is

$$
\alpha_b = 1 - 1/Z = 1 - \frac{S_S R_N}{S_N R_S} = \frac{b}{S_N}.
\tag{4}
$$

This is clearly an unrealistic model in several respects. First, it can be expected that there are SDMs in many populations and this will lead to an underestimation of $\alpha_b$, and second, it is likely that new balanced polymorphisms will be arising all the time and these will contribute to private polymorphism, increasing $R_N/R_S$ and leading to a conservative estimate of $\alpha_b$.

To investigate the extent to which this estimate might be biased we ran simulations, assuming that synonymous mutations were neutral and nonsynonymous mutations were deleterious, with their selection coefficients drawn from a gamma distribution; we simulated loci with and without a single balanced polymorphism in the centre of the locus. We then mixed these simulations and estimated $\alpha_b$ comparing it to the true value of $\alpha_b$. We considered 2 sampling points at 0.2 and 1.0 N generations after the populations had divided, where N is the ancestral population size. We find that $\alpha_b$ is almost always underestimated, and that the underestimation is greater for lower frequency polymorphisms (S16–S33 Figs); this is expected, since SDMs are expected to depress the estimate of $\alpha_b$. Among the highest frequency polymorphisms, $\alpha_b$ is quite well estimated when the true value of $\alpha_b > 0.3$; in these cases $\alpha_b$ is >0.5 of its true value. The estimate is greater using private polymorphisms from the population that is

larger. There is 1 circumstance in which $\alpha_b$ can be overestimated; this is where there has been a bottleneck and then expansion; in this case $\alpha_b$ is overestimated in the expanding population among the highest frequency polymorphisms. Surprisingly, this overestimation only affects cases in which there is at least some level of balancing selection; if we consider only simulations in which there is no balancing selection then Z < 1, and $\alpha_b$ is underestimated (S5 Fig).

## Single gene power

Our method is unlikely to have much power to detect balancing selection in single genes, because rather than leveraging the effects of balancing selection on patterns of linked polymorphism, our method simply looks for an excess of shared polymorphism; in fact, linkage confounds the signal of balancing selection in our method. This is in contrast to most other methods, which consider patterns of linked polymorphism and can have considerable power to detect balancing selection on single genes [6,7,9–11,13–15]. To investigate whether our method has any power to detect balancing selection in single genes, we simulated a locus with structure conforming to the average human gene, in which an ancestral population was split into 2 descendant populations. In half our simulations, we introduced a balanced polymorphism into each exon, and in the other simulations there was no balancing selection. We find that the distribution of Z values overlaps substantially for the simulations with and without balancing selection, independent of the sampling time point (S34 Fig). If we make the locus 10-fold larger in terms of the number of exons and introns, we find the distributions show less overlap, but the overlap remains considerable (S35 Fig). This analysis demonstrates that the method has little power for single genes, or even small collections of genes.

## Data analysis—Humans

We have shown that the method has the potential to detect balancing selection under realistic evolutionary models. We therefore applied our method to human data from the 1000 Genomes Project [22] focussing on 4 populations—Africans, Europeans, East Asians, and South Asians. We derived confidence intervals on our estimates of Z by bootstrapping the data by gene. The analysis of the individual populations shows a mixed picture (Fig 2); generally, comparisons involving African private polymorphisms show Z > 1 for polymorphisms at frequencies above 0.1; the results among the Asian and European populations are more erratic, and it is clear from the confidence intervals that we cannot reliably estimate Z for many frequency categories. In fact, for many frequency categories we do not have enough polymorphism data to estimate Z. As a consequence, we summed the data for all frequencies above 0.1. Here, a more consistent picture emerges with the data from at least 1 population in each comparison showing Z > 1. In the comparisons involving African private polymorphisms, Z is significantly greater than 1 for the comparisons involving the Asian populations and for the comparison between the African and non-African populations. It is worth noting that our simulations suggest that Z will tend to vary between populations which imply that in some comparisons Z can be less than 1 in 1 population but greater than 1 in another if there are modest levels of balancing selection.

If we estimate $\alpha_b$ in those comparisons in which Z is significantly greater than 1, we estimate that approximately 2% to 4% of the nonsynonymous shared polymorphisms between the African and other human populations are subject to balancing selection (Table 1). These estimates are likely to be underestimates because there will still be SDMs segregating in our data, even though we have removed the lowest frequency variants (see simulation results). The proportions suggest that at least 200 to 400 polymorphisms, which are shared between the African and other populations, are maintained by balancing selection (Table 1).

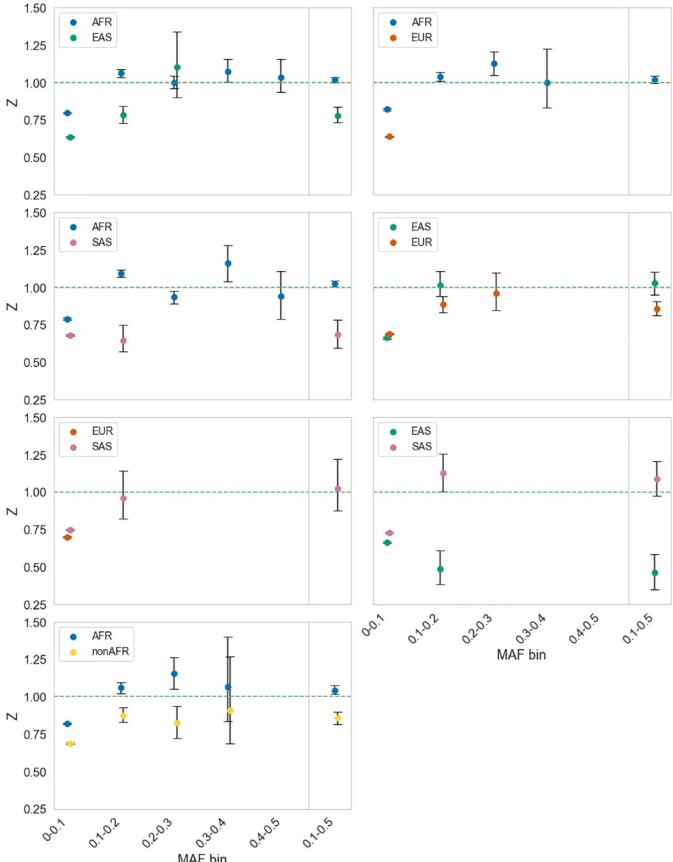

**Fig 2. Testing for balancing selection in human.** The value of Z is plotted against the frequency of shared and private polymorphisms, for pairs of populations: AFR, EAS, EUR, and SAS. In each panel, we show the value of Z for a comparison of 2 populations using the private polymorphisms from each, the population used being indicated in the plot legend. Data binned by minor allele frequency bins of size 0.1 on the x-axis. The final bin is 0.1–0.5 (i.e., all data minus the lowest frequency bin). Only data points in which there were at least 20 polymorphisms for all polymorphism categories were plotted, because the confidence intervals were very large otherwise. Code to extract and analyse the data can be found at https://github.com/vivaksoni/test_for_balancing_selection. The data underlying this figure can be found in S3 Data. AFR, Africans; EAS, East Asians; EUR, Europeans; SAS, South Asians.

A concern in any analysis of human population genetic data is the influence of biased gene conversion (BGC). This process tends to increase the number and allele frequencies of AT > GC mutations, and reduce the number and allele frequencies of GC > AT mutations. If this process differentially affects synonymous and nonsynonymous sites and shared and private polymorphisms, then it could potentially lead to Z > 1. To investigate whether BGC has

**Table 1. The level of balancing selection in humans.** Estimates of the proportion of shared nonsynonymous polymorphisms under balancing selection, $\alpha_b$, and the number of polymorphisms, $b$, being directly maintained by balancing selection for population comparisons in which Z > 1. Code to extract and analyse the data can be found at https://github.com/vivaksoni/test_for_balancing_selection.

| Target population | Comparative population | $\alpha_b$ | $\alpha_{b\_low}$ | $\alpha_{b\_high}$ | $b$ | $b_{low}$ | $b_{high}$ |
|---|---|---|---|---|---|---|---|
| African | Non-African | 0.0407 | 0.0123 | 0.0671 | 366 | 111 | 604 |
| African | European | 0.0400 | 0.0100 | 0.0600 | 577 | 176 | 926 |
| African | East Asian | 0.0174 | 0.0003 | 0.0351 | 223 | 4 | 451 |
| African | South Asian | 0.0251 | 0.0064 | 0.0439 | 341 | 87 | 595 |

**Table 2. Testing for the influence of biased gene conversion I.** Estimates of $Z$ for data split by median recombination rate. Code to extract and analyse the data can be found at https://github.com/vivaksoni/test_for_balancing_selection.

| Mean recombination rate | Z | $Z_{low}$ | $Z_{high}$ |
|---|---|---|---|
| $1.20 \times 10^{-9}$ | 1.02 | 0.99 | 1.06 |
| $1.80 \times 10^{-8}$ | 1.00 | 0.96 | 1.04 |

an effect, we performed 2 analyses. In the first, we divided our genes according to whether they were in high and low recombining regions, dividing the data at the median recombination rate (RR). Our 2 groups differ substantially in their mean rate of recombination (mean RR in low group = $1.2 \times 10^{-7}$ centimorgans per site and high group = $1.8 \times 10^{-6}$ centimorgans per site). We find that Z is actually higher in the low RR regions, although not significantly so (Table 2). However, neither estimate of Z is significantly greater than 1.

In the second test of the influence of BGC on the value of Z, we limited our analysis to mutations that are not affected by BGC—i.e., G<>C and A<>T mutations. This reduces our dataset by about 80%. As a consequence, we summed the data for all polymorphisms with frequencies >0.1. We find that our estimates are largely unchanged compared to when all polymorphisms are included, except in the case of the African-East Asian comparison; however, the confidence intervals are increased substantially so that Z is not significantly greater than 1 for any comparison (Table 3). Our 2 tests are inconclusive; in both cases, our values of Z are largely unaffected, but the reduction in sample size increases the variance of our estimate and all estimates become nonsignificant.

## Groups of genes

We can potentially apply our test of balancing selection to individual genes or groups of genes, where we have enough data. Balancing selection has been implicated in the evolution of immune-related genes (e.g., [4,15,23,24]), particularly major histocompatibility complex (MHC) or human leukocyte antigen (HLA) genes [25,26]. To investigate whether we could detect this signature in our data, we split our dataset into HLA and non-HLA genes [27]. Due to a lack of private polymorphisms, we combined all frequency categories >0.1. We find that Z > 1 for HLA genes in those population comparisons in which Z > 1 overall and in most cases this pattern is significant. We estimate that a very substantial proportion of nonsynonymous genetic variation is being maintained by balancing selection, although the confidence intervals on our estimates are large; roughly 50% of the shared nonsynonymous SNPs are being maintained by balancing selection between African and non-African populations in the HLA region and this equates to approximately 200 polymorphisms (Table 4). If we consider non-HLA genes, we find that Z > 1; however, the values are never significant and the

**Table 3. Testing for the effects of biased gene conversion II.** The values of Z when only G<>C and A<>T mutations are considered. Code to extract and analyse the data can be found at https://github.com/vivaksoni/test_for_balancing_selection.

| Target population | Comparative population | All polymorphism data | | | Filtered for BGC | | |
|---|---|---|---|---|---|---|---|
| | | Z | $Z_{low}$ | $Z_{high}$ | Z | $Z_{low}$ | $Z_{high}$ |
| African | Non-African | 1.04 | 1.01 | 1.07 | 1.03 | 0.94 | 1.12 |
| African | East Asian | 1.02 | 1.00 | 1.04 | 0.96 | 0.91 | 1.02 |
| African | South Asian | 1.03 | 1.01 | 1.05 | 1.02 | 0.96 | 1.08 |

BGC, biased gene conversion.

**Table 4. Balancing selection in HLA genes.** Estimates of the proportion of shared nonsynonymous polymorphisms under balancing selection, $\alpha_b$, and the number of polymorphisms being directly maintained by balancing selection, $b$, for population comparisons in the HLA region for population comparisons in which $Z > 1$ when using all genes. Estimates for polymorphisms with frequency >0.1. Missing values indicate the lower confidence interval was less than 1. Data consist of 177 genes. Code to extract and analyse the data can be found at https://github.com/vivaksoni/test_for_balancing_selection.

| Target | Comparative | $\alpha$ | $\alpha_{b\_low}$ | $\alpha_{b\_low}$ | $b$ | $b_{low}$ | $b_{high}$ |
|---|---|---|---|---|---|---|---|
| AFR | Non-AFR | 0.70 | 0.19 | 0.79 | 208 | 56 | 233 |
| AFR | EAS | 0.28 | - | 0.46 | 131 | - | 213 |
| AFR | SAS | 0.54 | 0.28 | 0.69 | 253 | 129 | 323 |

AFR, Africans; EAS, East Asians; HLA, human leukocyte antigen; SAS, South Asians.

**Table 5. Balancing selection in non-HLA genes.** Estimates of the proportion of shared nonsynonymous polymorphisms under balancing selection, $\alpha_b$, in non-HLA genes, and the number of polymorphisms being directly maintained by balancing selection, $b$, for population comparisons in which $Z > 1$ when using all genes. Missing values indicate the lower confidence interval was less than 1. Data consist of 19,212 genes. Code to extract and analyse the data can be found at https://github.com/vivaksoni/test_for_balancing_selection.

| Target | Comparative | $\alpha_b$ | $\alpha_{b\_low}$ | $\alpha_{b\_high}$ | $b$ | $b_{low}$ | $b_{high}$ |
|---|---|---|---|---|---|---|---|
| AFR | Non-AFR | 0.024 | - | 0.050 | 207 | - | 433 |
| AFR | EAS | 0.002 | - | 0.020 | 21 | - | 245 |
| AFR | SAS | 0.009 | - | 0.025 | 120 | - | 332 |

AFR, Africans; EAS, East Asians; HLA, human leukocyte antigen; SAS, South Asians.

estimated proportion of shared polymorphisms that are being maintained by balancing selection is very low (Table 5).

If we run our analysis grouping genes by their Gene Ontology (GO) category and restricting the analysis to those groups that have at least 100 polymorphisms with frequencies >0.1, we find 606 categories in which Z is significantly greater than 1 in at least 1 population comparison comparing all pairs of populations (S1 Fig). We list those significant in 5 or more population comparisons in Table 6. One of these GO categories, "endoplasmic reticulum membrane" is shared across 6 of the 14 population comparisons; among those categories shared among 5 are "viral process" and "response to stimulus." Fifty-four categories are shared between 4 or more population comparisons, and 108 among 3 or more population comparisons. These include 6 categories related to immunity (including immune system process which is significant in 5 population comparisons), and 40 categories that are linked to antigen presentation

**Table 6. GO category analysis.** GO categories in which Z is significantly greater than 1 in at least 5 population comparisons. Code to extract and analyse the data can be found at https://github.com/vivaksoni/test_for_balancing_selection.

| GO category | Counts |
|---|---|
| Endoplasmic reticulum membrane | 6 |
| Nucleic acid binding | 5 |
| Viral process | 5 |
| Response to stimulus | 5 |
| Intermediate filament | 5 |
| Zinc ion binding | 5 |
| Chromatin binding | 5 |
| Chromosome | 5 |

**Table 7. Single gene analysis.** Genes with Z > 1 in multiple population comparisons. Code to extract and analyse the data can be found at https://github.com/vivaksoni/test_for_balancing_selection.

| Gene symbol | Number of population comparisons in which Z > 1 |
|---|---|
| MUC4 | 14 |
| RP1L1 | 14 |
| PKD1L2 | 14 |
| ZAN | 14 |
| C1orf167 | 13 |
| SPTBN5 | 12 |
| MKI67 | 12 |
| DNAH14 | 11 |
| WDFY4 | 10 |
| FAM230G | 10 |
| CMYA5 | 9 |
| CRIPAK | 9 |
| SYNE2 | 9 |
| FSIP2 | 9 |
| GREB1 | 9 |
| ALMS1 | 9 |
| MUC19 | 9 |
| CENPF | 9 |

though not classified as immune-related categories. There are also 2 viral-related categories (including viral process which is significant in 5 population comparisons).

## Individual genes

Although our test is likely to have little power for individual genes (see above), we applied our statistic, combining all frequency bins (0 to 0.5) due to a lack of polymorphism data. We tested for significance using a 1-tailed Fisher's exact test. Of the 14,261 genes, we analysed 514 had Z > 1 in at least 1 population comparison. Eighteen of these were nominally significant at $p < 0.1$ (S2 Data), but no gene was individually significant when we corrected for multiple testing using a Bonferroni correction. Eighteen genes have Z > 1 in at least 9 population comparisons; note that since populations share polymorphisms, we cannot combine the evidence for balancing selection across these populations (Table 7). Four of these genes, MUC4, RP1L1, PKD1L2, and ZAN, have Z > 1 in all population comparisons.

If we use the 514 genes and do a GO enrichment analysis, we find multiple GO categories enriched for these genes including immune response categories with 3-fold enrichment. The most highly enriched categories are involved in energy production and conversion (including dynein binding) and intracellular transport (including microtubule motor activity) (S2 Data).

## Discussion

We propose a new method for detecting and quantifying the amount of balancing selection that is operating on polymorphisms, in which the numbers of nonsynonymous and synonymous polymorphisms that are shared between populations and species are compared to those that are private. The method is analogous to the McDonald–Kreitman test used to test and quantify the amount of adaptive evolution between species [16]. Our method is simple to apply and yields an estimate of the number of polymorphisms directly subject to balancing

selection, as opposed to those affected by linkage. We show that our test is robust to the presence of SDMs under simple demographic models of population division, expansion, and migration. When we apply our method to data from human populations, we find evidence that hundreds of nonsynonymous polymorphisms are probably being maintained by balancing selection in human populations. However, most of this signal comes from the HLA region.

Our method for detecting balancing selection appears to be robust to changes in demography. The classic MK test of adaptive evolution between species can generate artefactual evidence of adaptive evolution if there are SDMs and there has been population size expansion [16,28]; this is because SDMs that might have been fixed when the effective population size was small, no longer segregate once the population size is large. A similar bias does not appear to affect our test, although we have only investigated 2 DFEs and a limited number of demographic scenarios. Our test is likely to be more robust than the classic MK test because the shared polymorphisms are affected by the demographic changes that affect the private polymorphisms, i.e., if the population expands this will increase the effectiveness of natural selection on both the private and the shared polymorphisms. However, although our method seems to be relatively robust to changes in demography, in the sense that it does not generate artefactual evidence of balancing selection, it is evident that demography does affect the chance of balancing selection being identified, because the values of Z depend on the demography and which population the private polymorphisms are taken from (Fig 2). Furthermore, the method generally underestimates the number of balanced polymorphisms.

The method can in principle be applied to any pair of populations or species. However, the test is likely to be weak when the populations/species are closely related for 2 reasons. First, there will be relatively few private polymorphisms, and second, the proportion of shared polymorphisms that are subject to balancing selection is likely to be low, because so many neutral polymorphisms are shared between populations because of recent common ancestry. As the populations/species diverge so the number of private polymorphisms will increase, and the proportion of shared polymorphisms that are balanced will increase. Of course, as the time of divergence increases so the selective conditions that maintained the polymorphism are likely to change and the polymorphism might become neutral or subject to directional selection.

Our method is also likely, like all methods, to be better at detecting balanced polymorphisms that are common, because most populations are dominated by large numbers of rare neutral variants. The method requires that the neutral and selected sites are interdigitated; the method is therefore easy to apply to protein coding sequences, but may be more difficult to apply to other types of variation, such as that which affects gene expression. The method is weakly powered to detect balancing selection in individual genes (S34 and S35 Figs). Most other methods or analyses have leveraged patterns of variation in LD with a balanced polymorphism [6–15]; such variation obscures the signal that our method detects, which is an excess of shared variation.

The great advantage of our method is that it gives an estimate of the proportion and number of shared polymorphisms that are directly subject to balancing selection, under a set of simplifying assumptions, and it is simple to apply. However, the method is likely to yield underestimates of the proportion of balanced polymorphisms, under more realistic models of evolution, something we have confirmed by simulation (S16–S33 Figs). We have assumed, in deriving $\alpha_b$, that all nonsynonymous mutations are either strongly deleterious, neutral, or subject to balancing selection. However, a substantial fraction of nonsynonymous mutations appear to be slightly deleterious in humans [19,29–32] and other species [19,30,33,34]—i.e., they are deleterious, but sufficiently weakly selected that they contribute to polymorphism. Under stationary population size assumptions—i.e., in which the ancestral population is duplicated to form the daughter populations—this will lead to an underestimate of $\alpha_b$ because

SDMs tend to contribute more to private than shared polymorphism, and hence inflate $R_N/R_S$ relative to $S_N/S_S$ (Fig 1). Under more realistic demographic models, in which at least one of the derived populations is reduced, this is expected to depress $\alpha_b$ in the population that is being reduced because more SDMs will tend to segregate in smaller populations, hence inflating $R_N/R_S$ (compare Fig 2 and S3 Fig).

The second reason that we are likely underestimating the number of balanced polymorphisms using our simple method is that we assume that there are no balanced polymorphisms that are private to each population; these would inflate $R_N/R_S$. Private balanced polymorphisms might arise from an ancestral polymorphism that is lost from 1 of the daughter populations or 1 that arises de novo. A more realistic model of balancing selection is one in which balanced polymorphisms are continually generated with the selective forces persisting for some time before they dissipate [35] and the balanced polymorphism is lost. The process of population division itself is likely to lead to the loss of many balanced polymorphisms as the environment shifts in the 2 daughter populations.

A potential solution to the tendency for our method to underestimate Z and $\alpha_b$ is to simulate data under a realistic demographic model both with and without balancing selection, and use the simulations to estimate the proportion of balanced polymorphisms. However, there are challenges in this approach; in particular, we need an accurate demographic model. We have performed simulations under the commonly used human demographic model inferred by Gravel and colleagues [36] estimating the DFE from the current African population, assuming no balancing selection; we chose the African population because it has been subject to relatively modest demographic change. Our observed Z values do not match the simulated values (S36 Fig); in particular, we find that the observed values of Z are substantially greater than the simulated among the low frequency polymorphisms. However, the model of Gravel and colleagues does not fit the site frequency spectrum (SFS) of the individual populations of 1,000 genome data; for example, in the African population there are far too many singleton SNPs even among the putative neutral synonymous mutations (S37 Fig). The lack of fit is perhaps not surprising; Gravel and colleagues inferred their model using 80 chromosomes per population, whereas the 1,000 genome data contain >1,000 chromosomes per population. Furthermore, the inference of a demographic model should take into account the influence of BGC and background selection, which appear to be pervasive factors in the human genome [37], so these simulations will be complex.

We have analysed data from human populations and find some evidence for widespread balancing selection, particularly using private polymorphisms from the African population. It might be argued that detecting a signal of balancing selection using the private polymorphisms from 1 population is weak evidence of balancing selection. However, simulations suggest that this is likely to be common under many demographic models (S1–S15 Figs) when there are modest levels of balancing selection.

Controlling for BGC in our data analysis leads to inconclusive results; our estimates are not greatly affected by BGC, but because of the reduction in the sample size the confidence intervals increase and our estimates are not significantly different from zero. Much of the signal for balancing selection comes from the HLA genes. However, an analysis of GO categories suggests that numerous categories show evidence of balancing selection across multiple population comparisons (S1 Data). Some of these are expected, but many are not, such as "nucleic acid binding," which is significant in 5 of the 14 population comparisons (12 population comparisons plus African–non-African).

No individual gene is significant when we control for multiple testing; however, several genes have Z > 1 in multiple population comparisons including 10 that are shared across at least 10 of the 14 population comparisons. Three of these overlap with previous genome-wide

scans of selection, namely the protein-coding gene DNAH14, implicated in brain compression and encoding axonemal dynein [38]; MUC4, implicated in biliary tract cancer [39]; and ZAN, which encodes a protein involved in sperm adhesion, previously implicated in balancing selection and positive selection in human populations [40]. Two of these 10 genes are associated with tumours. MKI67 expression is associated with a higher tumour grade and early disease recurrence [41], and WDFY4 plays a critical role in the regulation of certain viral and tumour antigens in dendritic cells [42]. PKD1L2 is associated with polycystic kidney disease, and RP1L1 variants are associated with several retinal diseases including occult macular dystrophy [43]. SPTBN5 encodes for the cytoskeletal protein spectrin that plays a role in maintaining cytoskeletal structure [44], and C1orf167 expresses open reading frame protein that is highly expressed in the testis [45]. Finally, FAM230G is highly expressed in testes [46].

Twenty-five of the 514 genes with Z > 1 overlap with those genes identified by Bitarello and colleagues [15], but this is similar to the level of overlap expected at random, i.e., they observed that 7.9% of protein coding genes overlapped regions identified by their method as being subject to balancing selection, and we identified 514 candidates, so we expect $0.079 \times 514 = 41$ by chance alone. The lack of a significant overlap is possibly not surprising; we have applied our method to nonsynonymous variation, whereas the method of Bitarello and colleagues [15] considers all variation. Furthermore, the method of Bitarello and colleagues [15] is most powerful at detecting balancing selection over long time periods; in the case of humans, over periods of millions of years. In contrast, we have applied our method to populations that diverged 10,000s of years ago.

A signature of overdominance or heterozygous advantage can be produced by linkage to recessive or partially recessive deleterious mutations. For example, let us imagine that we have 2 closely linked loci at which we have deleterious alleles; let the A2 allele be the recessive allele at the A locus and the B2 allele at the B locus. Now consider a third neutral locus with alleles C1 and C2. If C1 is in LD with the A2 allele, and C2 is in LD with the B2 allele, then C1C2 heterozygous individuals will have higher fitness than C1C1 and C2C2 homozygotes. This form of selection is known as associative overdominance and can lead to the maintenance of genetic variation [47] in low RR regions. However, there is no reason why nonsynonymous mutations should be linked to other deleterious recessives more frequently than synonymous mutations, and Z is not substantially greater in regions of low recombination, so associative overdominance seems an unlikely explanation for our results (Table 2).

## Conclusion

We present a new approach to test for the presence of balancing selection and to the number of polymorphisms that are directly affected by it. Our method appears to be robust to demographic change. Application of the method to human population genetic data suggests that 100s of nonsynonymous polymorphisms shared between populations might be maintained by balancing selection.

## Methods and materials

### Human data

Human variation data were obtained from 1,000 genomes Grch37 vcf files [22]. Variants were annotated using Annovar's hg19 database [48]. The annotated data were then parsed to remove multinucleotide polymorphisms and indels. Because 1,000 genomes data provide allele frequencies for the non-reference allele rather than the minor allele, the minor allele frequency for each superpopulation and also for the global minor allele frequency was calculated. We used 1,000 genomes from the African, South Asian, East Asian, and European populations.

The American population was removed due to the fact that it is an admixed population. GO category information was obtained from Ensembl's BioMart data mining tool [18]. We used pyrho demography-aware recombination rate maps [49] for analyses that control for recombination rate.

## Data analysis

We calculated our test statistic Z for each pair of human populations, and also for the comparison between African and non-African data separating polymorphisms by frequency into bins of 0.1. We do not attempt to orient SNPs but use the folded site frequency spectrum. This is because there are potential difficulties with inferring the ancestral state when some sites such as CpG dinucleotides have rates of mutation; this is compounded by the fact that there is substantial variation in the mutation rate that is not associated with sequence context [50] and is therefore difficult to control for; as a consequence, a fraction of high frequency variants may simply be due to misinference. The folded site frequency spectrum does not suffer from these problems. We take the frequency of the shared polymorphism to be the frequency in the population from which the private polymorphisms are drawn. To test for statistical significance, we summed the values of $S_N$, $S_S$, $R_N$, and $R_S$ across genes and bootstrapped the data by gene 100 times to derive the 95% confidence intervals and standard error.

## Simulations

All simulations were run using the SLiM 3.1 [51]. Parameter values were taken from human estimates. Almost all simulations were of a 288 bp locus, this being the average size of a human exon [18]. Unless otherwise stated, the scaled recombination rate and scaled mutation rate were set at $r = 1.1 \times 10^{-8}$ [52], $\mu = 2.5 \times 10^{-8}$ [53] in the ancestral population. The distribution of fitness effects was assumed to be a gamma distribution, and the shape and mean strength of selection estimates for humans were taken from Eyre-Walker and colleagues [17] (shape parameter $\beta = 0.23$; mean $N_e s = 425$). For *Drosophila*, estimates were taken from Keightley and Eyre-Walker [54] ($\beta = 0.35$; mean $N_e s = 1,800$); again these were values in the ancestral population. Unless dominance was fixed, it was calculated using the model of Huber and colleagues [55], which was estimated from *Arabidopsis* species. The Huber model varies the dominance coefficient depending on the selection coefficient of the mutation, where the dominance coefficient increases with the strength of selection. Its formula is $h = f(s) = \frac{1}{\frac{1}{\theta_{intercept}} - \theta_{rate}s}$, where

$\theta_{intercept}$ defines the values of $h$ at $s = 0$, and $\theta_{rate}$ determines how quickly $h$ approaches 0 with decreasing negative selection coefficient. We set $\theta_{intercept}$ to 0.5 so that all mutations with a selection coefficient of $s = 0$ have a dominance coefficient, $h = 0.5$, and $\theta_{rate} = 41225.56$. This assumes an inverse relationship between $h$ and $s$, which gives the highest log likelihood score of the relationships compared by Huber and colleagues [55]. For balancing selection simulations, we assume a model of negative frequency-dependent selection; the equilibrium frequency was sampled from a uniform distribution between 0 and 1, with the $Ns$ value at equilibrium set to 20, where $N$ is the ancestral population size (see recipe 10.4.1 in SLiM [51] for details on how this was coded); however, it should be noted that some balanced polymorphisms with low equilibrium frequencies were lost in one of the descendent populations, so the realised distribution of frequencies is biased towards common polymorphisms (S38 Fig). Simulations in which the balanced polymorphism was lost from one of the 2 populations were discarded. The balanced polymorphism is introduced at the centre of the 288-bp region. Two million simulation runs were conducted for each model. This reduced the standard error on our estimates of Z to very low levels.

For the generic simulations (i.e., not those involving the human demographic model), the ancestral population size was set at 200. This was allowed to equilibrate for 15 N generations before a balanced polymorphism was introduced 5 N generations before the population was split into 2. The descendant populations were then sampled every 0.05 N generations up to 20 N generations after the split. We ran 5 different generic simulations: (i) simulations in which the ancestral population was duplicated; (ii) vicariance simulations in which the ancestral population was divided between the daughter populations in splits of 0.5 N to 0.5 N, 0.75 N to 0.25 N, 0.9 N to 0.1 N; (iii) variance simulations in which the descendant populations expanded; (iv) dispersal simulations, in which some variable fraction (0.5 N, 0.25 N, 0.1 N) of the ancestral population is duplicated to form the dispersal population, and the ancestral population continues as the other daughter population; and (v) dispersal with population increase of the dispersal population. The dispersal population starts as 0.1 N and expands exponentially 2 to 10× its original size after 21 N generations. Scenarios (ii) to (v) were repeated with migration rates of 0.01 N and 0.001 N of the ancestral population size between the descendant populations.

To investigate the power of the method to detect balancing selection in single genes, we ran a series of simulations of a single human gene; on average human genes are 32 kb in length, with an average exon size of 288 bp [18], 8.8 exons per gene, and 7.8 introns [56]. We simulated 9 exons of length 288 bp separated by 8 introns of 5,419 bp [56]. These loci were subject to human levels of mutation and recombination. We also ran a series of simulations of a gene that was 10-fold larger, in terms of the number of introns and exons. We ran simulations in which all mutations were deleterious and drawn from a gamma distribution, and a series of simulations in which a balanced polymorphism was introduced in the centre of each exon 5 N generations before the population was divided into 2 equal size populations (half the original population size). We only kept those balancing selection simulations in which at least 1 balance polymorphism survived to the sampling time point in both populations. In these simulations, we calculated Z using polymorphisms at all frequencies.

We also ran some simulations under the human demographic model of Gravel and colleagues [36]. The distribution of fitness effects for deleterious mutations was assumed to be a gamma distribution using the parameters estimated from the African superpopulation using the GammaZero model within the Grapes software [57]; the parameters are similar to those estimated by Eyre-Walker and colleagues [17], and used in the generic simulations (gamma shape = 0.17 and mean $N_e s$ = 1144). We chose to infer the DFE for the African superpopulation because this is currently the largest dataset available for a population that has been inferred to be relatively stable. Dominance was calculated using the Huber model discussed above. Sampling of all populations (African, East Asian, and European) was conducted at the end of the simulation (i.e., the equivalent of the present day). Each simulation was run 2 million times.

## Supporting information

**S1 Data. GO categories for which Z is significantly greater than 1, for each of the population comparisons.**
(XLSX)

**S2 Data. Individual genes for which Z > 1, for each of the population comparisons.**
(XLSX)

**S3 Data. Data underlying Fig 2 and S36–S38 Figs.**
(XLSX)

**S1 Fig. Vicariance simulations, with human DFE, in which the ancestral population splits to form 2 daughter populations of the size specified in the panel.** Each column is a separate set of simulations, with the top row plotting Z against tMRCA (measured in N generations, where N is the population size) for the larger daughter population, and the bottom row the smaller. There is no balancing selection and deleterious mutations are drawn from a gamma DFE with parameters inferred from human population data. Code to run these simulations can be found at https://github.com/vivaksoni/test_for_balancing_selection. DFE, distributions of fitness effect; tMRCA, time to the most recent common ancestor.
(TIF)

**S2 Fig. Dispersal simulations, with human DFE, in which a single daughter population disperses from the ancestral population.** Each column is a separate set of simulations, with the top row plotting Z against tMRCA (measured in N generations, where N is the population size) for the ancestral population, and the bottom row the daughter population. There is no balancing selection and deleterious mutations are drawn from a gamma DFE with parameters inferred from human population data. Code to run these simulations can be found at https://github.com/vivaksoni/test_for_balancing_selection. DFE, distributions of fitness effect; tMRCA, time to the most recent common ancestor.
(TIF)

**S3 Fig. Vicariance and expansion simulations, with human DFE, in which both daughter populations expand.** The ancestral population (of size $N = 200$) splits to form 2 daughter populations of size $N = 100$. Both daughter populations go on to expand in size. In the left column, the daughter populations double in size. In the right panel, they reach 10× their initial size. There is no balancing selection and deleterious mutations are drawn from a gamma DFE with parameters inferred from human population data. Code to run these simulations can be found at https://github.com/vivaksoni/test_for_balancing_selection. DFE, distributions of fitness effect; tMRCA, time to the most recent common ancestor.
(TIF)

**S4 Fig. Vicariance and expansion simulations, with human DFE, in which only 1 daughter population expands.** The ancestral population (of size $N = 200$) splits to form 2 daughter populations of size $N = 100$. One daughter population (upper panels) goes on to expand in size. In the left column, the daughter populations double in size. In the right panel, they reach 10× their initial size. There is no balancing selection and deleterious mutations are drawn from a gamma DFE with parameters inferred from human population data. Code to run these simulations can be found at https://github.com/vivaksoni/test_for_balancing_selection. DFE, distributions of fitness effect; tMRCA, time to the most recent common ancestor.
(TIF)

**S5 Fig. Dispersal and expansion simulations, with human DFE, in which a single daughter population disperses from the ancestral population and then expands.** The ancestral population (of size $N = 200$) splits to form a daughter population of size $N = 100$, which expands to the final population size shown in the panel. Each column is a separate set of simulations, with the top row plotting Z against tMRCA (measured in N generations, where N is the population size) for the ancestral population, and the bottom row the daughter population. There is no balancing selection and deleterious mutations are drawn from a gamma DFE with parameters inferred from human population data. Code to run these simulations can be found at https://github.com/vivaksoni/test_for_balancing_selection. DFE, distributions of fitness effect; tMRCA, time to the most recent common ancestor.
(TIF)

**S6 Fig. Vicariance simulations, with *Drosophila* DFE, in which the ancestral population splits to form 2 daughter populations of the size specified in the panel.** Each column is a separate set of simulations, with the top row plotting Z against tMRCA (measured in N generations, where N is the population size) for the larger daughter population, and the bottom row the smaller. There is no balancing selection and deleterious mutations are drawn from a gamma DFE with parameters inferred from *D. melanogaster* population data. Code to run these simulations can be found at https://github.com/vivaksoni/test_for_balancing_selection. DFE, distributions of fitness effect; tMRCA, time to the most recent common ancestor. (TIF)

**S7 Fig. Dispersal simulations, with *Drosophila* DFE, in which a single daughter population disperses from the ancestral population.** Each column is a separate set of simulations, with the top row plotting Z against tMRCA (measured in N generations, where N is the population size) for the ancestral population, and the bottom row the daughter population. There is no balancing selection and deleterious mutations are drawn from a gamma DFE with parameters inferred from *D. melanogaster* population data. Code to run these simulations can be found at https://github.com/vivaksoni/test_for_balancing_selection. DFE, distributions of fitness effect; tMRCA, time to the most recent common ancestor. (TIF)

**S8 Fig. Vicariance expansion simulations, with *Drosophila* DFE, in which both daughter populations expand.** The ancestral population (of size $N = 200$) splits to form 2 daughter populations of size $N = 100$. Both daughter populations go on to expand in size. In the left column, the daughter populations double in size. In the right panel, they reach 10× their initial size. There is no balancing selection and deleterious mutations are drawn from a gamma DFE with parameters inferred from *D. melanogaster* population data. Code to run these simulations can be found at https://github.com/vivaksoni/test_for_balancing_selection. DFE, distributions of fitness effect; tMRCA, time to the most recent common ancestor. (TIF)

**S9 Fig. Vicariance expansion simulations, with *Drosophila* DFE, in which only 1 daughter population expands.** The ancestral population (of size $N = 200$) splits to form 2 daughter populations of size $N = 100$. One daughter population (upper panels) goes on to expand in size. In the left column, the daughter populations double in size. In the right panel, they reach 10× their initial size. There is no balancing selection and deleterious mutations are drawn from a gamma DFE with parameters inferred from *D. melanogaster* population data. Code to run these simulations can be found at https://github.com/vivaksoni/test_for_balancing_selection. DFE, distributions of fitness effect; tMRCA, time to the most recent common ancestor. (TIF)

**S10 Fig. Dispersal expansion simulations, with *Drosophila* DFE, in which a single daughter population disperses from the ancestral population and then expands.** The ancestral population (of size $N = 200$) splits to form a daughter population of size $N = 100$, which expands to the final population size shown in the panel. Each column is a separate set of simulations, with the top row plotting Z against tMRCA (measured in N generations, where N is the population size) for the ancestral population, and the bottom row the daughter population. There is no balancing selection and deleterious mutations are drawn from a gamma DFE with parameters inferred from *D. melanogaster* population data. Code to run these simulations can be found at https://github.com/vivaksoni/test_for_balancing_selection. DFE, distributions of fitness effect; tMRCA, time to the most recent common ancestor. (TIF)

**S11 Fig. Vicariance simulations with migration and human DFE, in which the ancestral population splits to form 2 daughter populations of the size specified in the panel.** Each column is a separate set of simulations, with the top row plotting Z against tMRCA (measured in N generations, where N is the population size) for the larger daughter population, and the bottom row the smaller. There is no balancing selection and deleterious mutations are drawn from a gamma DFE with parameters inferred from human population data. Migration rate is 0.01 N. Code to run these simulations can be found at https://github.com/vivaksoni/test_for_balancing_selection. DFE, distributions of fitness effect; tMRCA, time to the most recent common ancestor.
(TIF)

**S12 Fig. Dispersal simulations with migration and human DFE, in which a single daughter population disperses from the ancestral population.** Each column is a separate set of simulations, with the top row plotting Z against tMRCA (measured in N generations, where N is the population size) for the ancestral population, and the bottom row the daughter population. There is no balancing selection and deleterious mutations are drawn from a gamma DFE with parameters inferred from human population data. Migration rate is 0.01 N. Code to run these simulations can be found at https://github.com/vivaksoni/test_for_balancing_selection. DFE, distributions of fitness effect; tMRCA, time to the most recent common ancestor.
(TIF)

**S13 Fig. Dispersal expansion simulations with migration and human DFE, in which a single daughter population disperses from the ancestral population and then expands.** The ancestral population (of size $N = 200$) splits to form a daughter population of size $N = 100$, which expands to the final population size shown in the panel. Each column is a separate set of simulations, with the top row plotting Z against tMRCA (measured in N generations, where N is the population size) for the ancestral population, and the bottom row the daughter population. There is no balancing selection and deleterious mutations are drawn from a gamma DFE with parameters inferred from human population data. Migration rate is 0.01 N. Code to run these simulations can be found at https://github.com/vivaksoni/test_for_balancing_selection. DFE, distributions of fitness effect; tMRCA, time to the most recent common ancestor.
(TIF)

**S14 Fig. Simulations, with human DFE, for combined 0.1–0.5 minor allele frequencies.** Each panel is a separate simulated scenario, with population sizes listed in the panel legend. (*) indicates simulations with migration (with migration rate 0.01 N). The first number is for the filled in data lines, denoting the ancestral population in dispersal scenarios, and for the larger population in the vicariance scenarios. The second number is for the dotted data lines, denoting the daughter population in dispersal scenarios, and the smaller population in the vicariance scenarios. For more details on each scenario, please see S1–S10 Figs. There is no balancing selection and deleterious mutations are drawn from a gamma DFE with parameters inferred from human population data. Code to run these simulations can be found at https://github.com/vivaksoni/test_for_balancing_selection. DFE, distributions of fitness effect; tMRCA, time to the most recent common ancestor.
(TIF)

**S15 Fig. Simulations, with *Drosophila* DFE, for combined 0.1–0.5 minor allele frequencies.** Each panel is a separate simulated scenario, with population sizes listed in the panel legend. (*) indicates simulations with migration (with migration rate 0.01 N). The first number is for the filled in data lines, denoting the ancestral population in dispersal scenarios, and for the larger

population in the vicariance scenarios. The second number is for the dotted data lines, denoting the daughter population in dispersal scenarios, and the smaller population in the vicariance scenarios. For more details on each scenario, please see Supporting information S1–S10 Figs. There is no balancing selection and deleterious mutations are drawn from a gamma DFE with parameters inferred from *D. melanogaster* population data. Code to run these simulations can be found at https://github.com/vivaksoni/test_for_balancing_selection. DFE, distributions of fitness effect; tMRCA, time to the most recent common ancestor.
(TIF)

**S16 Fig. Comparison of $\alpha_{b\ inferred}$ and $\alpha_{b\ true}$ for dispersal simulation, sampled at 0.2 N generations after the population split, in which the ancestral population is of size $N$ = 200, and the daughter population is size $N$ = 20.** The top row plots are for the ancestral population, the bottom row for the daughter population. The left column plots $\alpha_{b\ inferred}/\alpha_{b\ true}$ against the proportion of balancing selection simulations. The right column plots $\alpha_{b\ true}$ against the proportion of balancing selection simulations. The 0–0.1 MAF category has been removed, and negative values have been truncated to 0 for the sake of clarity. Deleterious mutations are drawn from a gamma DFE with parameters inferred from human population data. Code to run these simulations can be found at https://github.com/vivaksoni/test_for_balancing_selection. DFE, distributions of fitness effect; MAF, minor allele frequency.
(TIF)

**S17 Fig. Comparison of $\alpha_{b\ inferred}$ and $\alpha_{b\ true}$ for dispersal simulation, sampled at 1 N generations after the population split, in which the ancestral population is of size $N$ = 200, and the daughter population is size $N$ = 20.** The top row plots are for the ancestral population, the bottom row for the daughter population. The left column plots $\alpha_{b\ inferred}/\alpha_{b\ true}$ against the proportion of balancing selection simulations. The right column plots $\alpha_{b\ true}$ against the proportion of balancing selection simulations. The 0–0.1 MAF category has been removed, and negative values have been truncated to 0 for the sake of clarity. Deleterious mutations are drawn from a gamma DFE with parameters inferred from human population data. Code to run these simulations can be found at https://github.com/vivaksoni/test_for_balancing_selection. DFE, distributions of fitness effect; MAF, minor allele frequency.
(TIF)

**S18 Fig. Comparison of $\alpha_{b\ inferred}$ and $\alpha_{b\ true}$ for dispersal simulation, sampled at 0.2 N generations after the population split, in which the ancestral population is of size $N$ = 200, and the daughter population is size $N$ = 50.** The top row plots are for the ancestral population, the bottom row for the daughter population. The left column plots $\alpha_{b\ inferred}/\alpha_{b\ true}$ against the proportion of balancing selection simulations. The right column plots $\alpha_{b\ true}$ against the proportion of balancing selection simulations. The 0–0.1 MAF category has been removed, and negative values have been truncated to 0 for the sake of clarity. Deleterious mutations are drawn from a gamma DFE with parameters inferred from human population data. Code to run these simulations can be found at https://github.com/vivaksoni/test_for_balancing_selection. DFE, distributions of fitness effect; MAF, minor allele frequency.
(TIF)

**S19 Fig. Comparison of $\alpha_{b\ inferred}$ and $\alpha_{b\ true}$ for dispersal simulation, sampled at 1 N generations after the population split, in which the ancestral population is of size $N$ = 200, and the daughter population is size $N$ = 50.** The top row plots are for the ancestral population, the bottom row for the daughter population. The left column plots $\alpha_{b\ inferred}/\alpha_{b\ true}$ against the proportion of balancing selection simulations. The right column plots $\alpha_{b\ true}$ against the

proportion of balancing selection simulations. The 0–0.1 MAF category has been removed, and negative values have been truncated to 0 for the sake of clarity. Deleterious mutations are drawn from a gamma DFE with parameters inferred from human population data. Code to run these simulations can be found at https://github.com/vivaksoni/test_for_balancing_selection. DFE, distributions of fitness effect; MAF, minor allele frequency.
(TIF)

**S20 Fig. Comparison of $\alpha_{b\ inferred}$ and $\alpha_{b\ true}$ for dispersal simulation, sampled at 0.2 N generations after the population split, in which the ancestral population is of size $N = 200$, and the daughter population is size $N = 100$.** The top row plots are for the ancestral population, the bottom row for the daughter population. The left column plots $\alpha_{b\ inferred}/\alpha_{b\ true}$ against the proportion of balancing selection simulations. The right column plots $\alpha_{b\ true}$ against the proportion of balancing selection simulations. The 0–0.1 MAF category has been removed, and negative values have been truncated to 0 for the sake of clarity. Deleterious mutations are drawn from a gamma DFE with parameters inferred from human population data. Code to run these simulations can be found at https://github.com/vivaksoni/test_for_balancing_selection. DFE, distributions of fitness effect; MAF, minor allele frequency.
(TIF)

**S21 Fig. Comparison of $\alpha_{b\ inferred}$ and $\alpha_{b\ true}$ for dispersal simulation, sampled at 1 N generations after the population split, in which the ancestral population is of size $N = 200$, and the daughter population is size $N = 100$.** The top row plots are for the ancestral population, the bottom row for the daughter population. The left column plots $\alpha_{b\ inferred}/\alpha_{b\ true}$ against the proportion of balancing selection simulations. The right column plots $\alpha_{b\ true}$ against the proportion of balancing selection simulations. The 0–0.1 MAF category has been removed, and negative values have been truncated to 0 for the sake of clarity. Deleterious mutations are drawn from a gamma DFE with parameters inferred from human population data. Code to run these simulations can be found at https://github.com/vivaksoni/test_for_balancing_selection. DFE, distributions of fitness effect; MAF, minor allele frequency.
(TIF)

**S22 Fig. Comparison of $\alpha_{b\ inferred}$ and $\alpha_{b\ true}$ for vicariance simulation, sampled at 0.2 N generations after the population split, in which the ancestral population is of size $N = 200$, and both daughter populations are size $N = 100$.** The top row plots are for 1 daughter population, the bottom row for the other. The left column plots $\alpha_{b\ inferred}/\alpha_{b\ true}$ against the proportion of balancing selection simulations. The right column plots $\alpha_{b\ true}$ against the proportion of balancing selection simulations. The 0–0.1 MAF category has been removed, and negative values have been truncated to 0 for the sake of clarity. Deleterious mutations are drawn from a gamma DFE with parameters inferred from human population data. Code to run these simulations can be found at https://github.com/vivaksoni/test_for_balancing_selection. DFE, distributions of fitness effect; MAF, minor allele frequency.
(TIF)

**S23 Fig. Comparison of $\alpha_{b\ inferred}$ and $\alpha_{b\ true}$ for vicariance simulation, sampled at 1 N generations after the population split, in which the ancestral population is of size $N = 200$, and both daughter populations are size $N = 100$.** The top row plots are for 1 daughter population, the bottom row for the other. The left column plots $\alpha_{b\ inferred}/\alpha_{b\ true}$ against the proportion of balancing selection simulations. The right column plots $\alpha_{b\ true}$ against the proportion of balancing selection simulations. The 0–0.1 MAF category has been removed, and negative values have been truncated to 0 for the sake of clarity. Deleterious mutations are drawn from a gamma DFE with parameters inferred from human population data. Code to run these

simulations can be found at https://github.com/vivaksoni/test_for_balancing_selection. DFE, distributions of fitness effect; MAF, minor allele frequency.
(TIF)

**S24 Fig. Comparison of $\alpha_{b\ inferred}$ and $\alpha_{b\ true}$ for vicariance simulation, sampled at 0.2 N generations after the population split, in which the ancestral population is of size $N = 200$, with 1 daughter population size $N = 150$ (top row) and the other size $N = 50$ (bottom row).** The left column plots $\alpha_{b\ inferred}/\alpha_{b\ true}$ against the proportion of balancing selection simulations. The right column plots $\alpha_{b\ true}$ against the proportion of balancing selection simulations. The 0–0.1 MAF category has been removed, and negative values have been truncated to 0 for the sake of clarity. Deleterious mutations are drawn from a gamma DFE with parameters inferred from human population data. Code to run these simulations can be found at https://github.com/vivaksoni/test_for_balancing_selection. DFE, distributions of fitness effect; MAF, minor allele frequency.
(TIF)

**S25 Fig. Comparison of $\alpha_{b\ inferred}$ and $\alpha_{b\ true}$ for vicariance simulation, sampled at 1 N generations after the population split, in which the ancestral population is of size $N = 200$, with 1 daughter population size $N = 150$ (top row) and the other size $N = 50$ (bottom row).** The left column plots $\alpha_{b\ inferred}/\alpha_{b\ true}$ against the proportion of balancing selection simulations. The right column plots $\alpha_{b\ true}$ against the proportion of balancing selection simulations. The 0–0.1 MAF category has been removed, and negative values have been truncated to 0 for the sake of clarity. Deleterious mutations are drawn from a gamma DFE with parameters inferred from human population data. Code to run these simulations can be found at https://github.com/vivaksoni/test_for_balancing_selection. DFE, distributions of fitness effect; MAF, minor allele frequency.
(TIF)

**S26 Fig. Comparison of $\alpha_{b\ inferred}$ and $\alpha_{b\ true}$ for vicariance simulation, sampled at 0.2 N generations after the population split, in which the ancestral population is of size $N = 200$, with 1 daughter population size $N = 180$ (top row) and the other size $N = 20$ (bottom row).** The left column plots $\alpha_{b\ inferred}/\alpha_{b\ true}$ against the proportion of balancing selection simulations. The right column plots $\alpha_{b\ true}$ against the proportion of balancing selection simulations. The 0–0.1 MAF category has been removed, and negative values have been truncated to 0 for the sake of clarity. Deleterious mutations are drawn from a gamma DFE with parameters inferred from human population data. Code to run these simulations can be found at https://github.com/vivaksoni/test_for_balancing_selection. DFE, distributions of fitness effect; MAF, minor allele frequency.
(TIF)

**S27 Fig. Comparison of $\alpha_{b\ inferred}$ and $\alpha_{b\ true}$ for vicariance simulation, sampled at 1 N generations after the population split, in which the ancestral population is of size $N = 200$, with 1 daughter population size $N = 180$ (top row) and the other size $N = 20$ (bottom row).** The left column plots $\alpha_{b\ inferred}/\alpha_{b\ true}$ against the proportion of balancing selection simulations. The right column plots $\alpha_{b\ true}$ against the proportion of balancing selection simulations. The 0–0.1 MAF category has been removed, and negative values have been truncated to 0 for the sake of clarity. Deleterious mutations are drawn from a gamma DFE with parameters inferred from human population data. Code to run these simulations can be found at https://github.com/vivaksoni/test_for_balancing_selection. DFE, distributions of fitness effect; MAF, minor allele frequency.
(TIF)

**S28 Fig. Comparison of $\alpha_{b\ inferred}$ and $\alpha_{b\ true}$ for expansion simulation, sampled at 0.2 N generations after the population split, in which the ancestral population is of size $N$ = 200, and the daughter population is size $N$ = 40.** The top row plots are for the ancestral population, the bottom row for the daughter population. The left column plots $\alpha_{b\ inferred}/\alpha_{b\ true}$ against the proportion of balancing selection simulations. The right column plots $\alpha_{b\ true}$ against the proportion of balancing selection simulations. The 0–0.1 MAF category has been removed, and negative values have been truncated to 0 for the sake of clarity. Deleterious mutations are drawn from a gamma DFE with parameters inferred from human population data. Code to run these simulations can be found at https://github.com/vivaksoni/test_for_balancing_selection. DFE, distributions of fitness effect; MAF, minor allele frequency. (TIF)

**S29 Fig. Comparison of $\alpha_{b\ inferred}$ and $\alpha_{b\ true}$ for expansion simulation, sampled at 1 N generations after the population split, in which the ancestral population is of size $N$ = 200, and the daughter population is size $N$ = 40.** The top row plots are for the ancestral population, the bottom row for the daughter population. The left column plots $\alpha_{b\ inferred}/\alpha_{b\ true}$ against the proportion of balancing selection simulations. The right column plots $\alpha_{b\ true}$ against the proportion of balancing selection simulations. The 0–0.1 MAF category has been removed, and negative values have been truncated to 0 for the sake of clarity. Deleterious mutations are drawn from a gamma DFE with parameters inferred from human population data. Code to run these simulations can be found at https://github.com/vivaksoni/test_for_balancing_selection. DFE, distributions of fitness effect; MAF, minor allele frequency. (TIF)

**S30 Fig. Comparison of $\alpha_{b\ inferred}$ and $\alpha_{b\ true}$ for expansion simulation, sampled at 0.2 N generations after the population split, in which the ancestral population is of size $N$ = 200, and the daughter population is size $N$ = 400.** The top row plots are for the ancestral population, the bottom row for the daughter population. The left column plots $\alpha_{b\ inferred}/\alpha_{b\ true}$ against the proportion of balancing selection simulations. The right column plots $\alpha_{b\ true}$ against the proportion of balancing selection simulations. The 0–0.1 MAF category has been removed, and negative values have been truncated to 0 for the sake of clarity. Deleterious mutations are drawn from a gamma DFE with parameters inferred from human population data. Code to run these simulations can be found at https://github.com/vivaksoni/test_for_balancing_selection. DFE, distributions of fitness effect; MAF, minor allele frequency. (TIF)

**S31 Fig. Comparison of $\alpha_{b\ inferred}$ and $\alpha_{b\ true}$ for expansion simulation, sampled at 1 N generations after the population split, in which the ancestral population is of size $N$ = 200, and the daughter population is size $N$ = 400.** The top row plots are for the ancestral population, the bottom row for the daughter population. The left column plots $\alpha_{b\ inferred}/\alpha_{b\ true}$ against the proportion of balancing selection simulations. The right column plots $\alpha_{b\ true}$ against the proportion of balancing selection simulations. The 0–0.1 MAF category has been removed, and negative values have been truncated to 0 for the sake of clarity. Deleterious mutations are drawn from a gamma DFE with parameters inferred from human population data. Code to run these simulations can be found at https://github.com/vivaksoni/test_for_balancing_selection. DFE, distributions of fitness effect; MAF, minor allele frequency. (TIF)

**S32 Fig. Comparison of $\alpha_{b\ inferred}$ and $\alpha_{b\ true}$ for expansion simulation, sampled at 0.2 N generations after the population split, in which the ancestral population is of size $N$ = 200, and the daughter population is size $N$ = 2,430.** The top row plots are for the ancestral

population, the bottom row for the daughter population. The left column plots $\alpha_{b\ inferred}/\alpha_{b\ true}$ against the proportion of balancing selection simulations. The right column plots $\alpha_{b\ true}$ against the proportion of balancing selection simulations. The 0–0.1 MAF category has been removed, and negative values have been truncated to 0 for the sake of clarity. Deleterious mutations are drawn from a gamma DFE with parameters inferred from human population data. Code to run these simulations can be found at https://github.com/vivaksoni/test_ for_balancing_selection. DFE, distributions of fitness effect; MAF, minor allele frequency.
(TIF)

**S33 Fig. Comparison of $\alpha_{b\ inferred}$ and $\alpha_{b\ true}$ for expansion simulation, sampled at 1 N generations after the population split, in which the ancestral population is of size $N = 200$, and the daughter population is size $N = 2,430$.** The top row plots are for the ancestral population, the bottom row for the daughter population. The left column plots $\alpha_{b\ inferred}/\alpha_{b\ true}$ against the proportion of balancing selection simulations. The right column plots $\alpha_{b\ true}$ against the proportion of balancing selection simulations. The 0–0.1 MAF category has been removed, and negative values have been truncated to 0 for the sake of clarity. Deleterious mutations are drawn from a gamma DFE with parameters inferred from human population data. Code to run these simulations can be found at https://github.com/vivaksoni/test_for_balancing_ selection. DFE, distributions of fitness effect; MAF, minor allele frequency.
(TIF)

**S34 Fig. The distribution of Z for simulations with (orange) and without (blue) balancing selection for a locus that has average human dimensions.** For each scenario 500,000 simulations were run. (*** $p < 0.001$ for a test between 2 distributions). Code to run these simulations can be found at https://github.com/vivaksoni/test_for_balancing_selection.
(TIF)

**S35 Fig. The distribution of Z for simulations with (orange) and without (blue) balancing selection for a locus that is 10-fold larger than the average human gene.** For each scenario 500,000 simulations were run. (*** $p < 0.001$ for a test between 2 distributions). Code to run these simulations can be found at https://github.com/vivaksoni/test_for_balancing_selection.
(TIF)

**S36 Fig. Simulations using the Gravel model of human demography (Gravel and colleagues, 2011).** Shown are the observed (filled circles) and simulated (crosses) values of Z. Each column represents a different population comparison. From left to right: AFR and EAS, AFR and EUR, EUR and EAS. The population name in the upper left indicates which set of private polymorphisms are used to calculate Z in each population comparison. The x-axis represents private polymorphism minor allele frequency bins. Confidence intervals generated by bootstrapping. Code to extract and analyse the data can be found at https://github.com/ vivaksoni/test_for_balancing_selection. The data underlying this figure can be found in S3 Data. AFR, Africans; EAS, East Asians; EUR, Europeans.
(TIF)

**S37 Fig. Comparison of simulated (under the Gravel and colleagues (2011) model of human demography) and observed SFS from the African population.** The SFS is summarised by combining SNPs at counts of 2 and 3, 4 to 7, 8 to 15. . .etc. with singletons considered by themselves. Code to extract and analyse the data can be found at https://github.com/ vivaksoni/test_for_balancing_selection. The data underlying this figure can be found in S3 Data.
(TIF)

**S38 Fig. Balanced polymorphisms were introduced under a model of frequency-dependent selection in which the equilibrium frequency was drawn from a uniform distribution.** However, rare polymorphisms are more likely to be lost; the figure shows the average minor allele frequency of shared balanced polymorphisms in a simulation in which the population was duplicated and sampled N generations after the duplication event. Code to run these simulations can be found at https://github.com/vivaksoni/test_for_balancing_selection. The data underlying this figure can be found in S3 Data.
(TIF)

## Author Contributions

**Conceptualization:** Michiel Vos, Adam Eyre-Walker.

**Data curation:** Vivak Soni.

**Formal analysis:** Vivak Soni.

**Funding acquisition:** Michiel Vos.

**Investigation:** Vivak Soni, Adam Eyre-Walker.

**Methodology:** Vivak Soni, Adam Eyre-Walker.

**Project administration:** Adam Eyre-Walker.

**Supervision:** Adam Eyre-Walker.

**Validation:** Vivak Soni.

**Visualization:** Vivak Soni.

**Writing – original draft:** Vivak Soni, Adam Eyre-Walker.

**Writing – review & editing:** Vivak Soni, Michiel Vos, Adam Eyre-Walker.

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
