## [Editor Report · Decision Letter 0]

19 Feb 2021

Dear Adam, 

Thank you for submitting your manuscript entitled "A new test demonstrates that balancing selection maintains hundreds of non-synonymous polymorphisms in the human genome" for consideration as a Research Article by PLOS Biology.

Your manuscript has now been evaluated by the PLOS Biology editorial staff, as well as by an academic editor with relevant expertise, and I'm writing to let you know that we would like to send your submission out for external peer review.

Please re-submit your manuscript within two working days, i.e. by Feb 22 2021 11:59PM.

Kind regards,

Roli

Senior Editor

PLOS Biology

---

## [Decision Letter · Decision Letter 1]

12 Apr 2021

Dear Adam,

Thank you very much for submitting your manuscript "A new test demonstrates that balancing selection maintains hundreds of non-synonymous polymorphisms in the human genome" for consideration as a Research Article at PLOS Biology. Your manuscript has been evaluated by the PLOS Biology editors, an Academic Editor with relevant expertise, and by four independent reviewers.

You'll see that although the reviewers are broadly positive, some of them raise some significant concerns that will need to be addressed before further consideration. In particular, reviewers #2 and #3 make some strikingly concordant requests for re-structuring and a formal statistical framework, and reviewer #4 raises a number of potential sources of artefact.

In light of the reviews (below), we will not be able to accept the current version of the manuscript, but we would welcome re-submission of a much-revised version that takes into account the reviewers' comments. We cannot make any decision about publication until we have seen the revised manuscript and your response to the reviewers' comments. Your revised manuscript is also likely to be sent for further evaluation by the reviewers.

We expect to receive your revised manuscript within 3 months. 

**IMPORTANT - SUBMITTING YOUR REVISION**

*Re-submission Checklist*

*Published Peer Review*

*PLOS Data Policy*

*Blot and Gel Data Policy*

Best wishes,

Roli

Senior Editor,

rroberts@plos.org,

PLOS Biology

REVIEWERS' COMMENTS:

Reviewer #1:

In this manuscript, the authors present a new method for detecting balancing selection and estimating the fraction of polymorphisms maintained by balancing selection. The method inspired by the McDonald-Kreitman test derives statistic Z, which compares shared and private polymorphisms between two populations to estimate an excess fraction of shared alleles that are maintained by selection. The test faces some interpretational challenges, similar to the MK test, such as the influence of deleterious mutations, especially on the low-frequency private variants, but also the impact of population-specific balancing selection on the statistic. These aspects are investigated either with simulations, or they are thoroughly discussed. I think the biggest weakness is uncertainty about the actual number of variants under selection, given that this number can vary a lot depending on the population size and number of deleterious alleles. Nevertheless, the analyses suggest that if anything, the confounding factors underestimate balancing selection, but do not generate false positives. I feel that this method brings an interesting addition to the existing methods and adds new evidence to the view that balancing selection can be quite prevalent in the genome. 

Here I have two minor general comments and a few small specific comments to the text.

Estimated numbers of variants segregating under balancing selection are large approximations, given that Z and alpha can vary considerably depending on the population size, likely because of the prevalence of deleterious variants. On top of that, it's not clear what is the effect of the strength of balancing selection and the number of balanced polymorphisms on the estimate of alpha and the estimated number of variants under balancing selection. I was wondering how well the alpha estimates obtained in simulations relate to the true number of simulated balanced polymorphisms (=1) and how precisely can this method reflect the number of balanced polymorphisms in the populations.

Authors performed simulations with migration but they are not commented on in the manuscript and the results are not shown.

Line 174: In the legend description of subplots is unclear, I would suggest skipping "a) - orange" and "a) - green".

line 347: Please include the number of HLA and non-HLA genes that were used in the analysis in the text or the table legend.

line 350: The referenced figure (Fig. S13) is not the right one.

line 384: I think it should be: at least "one" population comparison.

line 384: Could you explain which populations are being compared here?

line 496: I think it should be: model "of" balancing selection.

line 513: I believe the reference should be figure S13 and S14 instead of S14 and S15.

line 633: Please include the version of the program.

line 637: Semicolon in superscript after the first citation.

Supplementary Tables 2 and 3 are not referenced in the text, and Supplementary Table S1 is redundant with Table 3.

To meet the requirements of the journal, please submit data or code to replicate the study.

Reviewer #2:

The MacDonald-Kreitman (MK) test has played a major role in molecular evolution studies. In the present paper the authors proposed a test similar in spirit to the MK test for balancing selection. Like the MK test it requires two types of sites that can be classified in two different ways. As in the MK test the two types of sites are generally synonymous and non synonymous changes. One assumes that the former evolve neutrally and the latter are under selection. The novelty in the present paper is to then consider polymorphisms that are shared between populations and polymorphisms that are private to one of the populations. So for the first class we will have S_S and S_N for synonymous and nonsynonymous changes at shared polymorphisms, respectively, and at the second one, we will have R_S and R_N. The authors then reasoned that under balancing selection one will have:

Z=S_N/S_S/R_N/R_S >1

They then went on to show that an estimate of the proportion of non-synonymous mutations subject directly to balancing selection is

alpha_b=1-S_S*R_N/S_N*R_S

There is no question that this is a neat and clever idea and I enjoyed reading the paper. Most of the paper is devoted to simulations to test factors influencing Z and alpha_b and to an application to human data. My main concerns are (i) that the simulations are presented in a way favourable to the test even though there is generally a caveat at the end of each paragraph stressing the limitations and (ii) the lack of an explicit statistical framework. I would have liked the paper to be divided in three parts: (i) simulations to investigate the behavior of Z and alpha_b, (ii) development of a statistical test at the genome level and at individual genes level, (iii) application to a human data set. The second part is simply completely lacking. Also, if possible it could be good to apply the method to a couple of other species with different life history traits and demographies. It could also be good to clarify what the Z statistics or alpha_b mean at the genome level vs individual gene level. What does it mean that evidence of balancing selection is observed globally but not at single loci? Is it simply a statistical power issue or could it mean something else?

Minor comments:

lines 209-210: why not call Watterson coefficient a_n as usual or give it explicitly?

line 258-263: the statement feels a bit contrived.

Line 496-97 A more realistic model OF balancing selection

Reviewer #3:

The manuscript of Soni et al describes a statistic to measure balancing selection in recently diverged populations inspired by the McDonald-Kreitman test. With some assumption, they present a derivation that uses this statistic to estimate the total number of alleles that are subject to balancing selection between two populations. Simulated data from two-population models were used to evaluate the measurement under several two-population scenarios. Subsequently, the statistic was applied to human genomics data using pairs of 1000 Genomes continental population groupings. The authors report evidence of balancing selection in these populations, particularly in continental African population groups relative to others. Finally, using GO-enrichment categories, gene groups were analyzed separately to identify specific types of genes with evidence of balancing selection. Where sample size allowed, these analyses were further narrowed down to identify individual genes with variants under balancing selection.

This manuscript proposes a statistic with two novel implications: (i) a measure of balancing selection better suited to detection on smaller time scales (thousands of years, rather than millions of years), and (ii) a direct estimate of the number of loci maintained by balancing selection in a given region between two populations. Either improvement could easily complement existing methods and approaches that are available (and the authors cite). However, with the analyses presented, the authors do not provide sufficient evidence supporting either of these two inferences. Specifically, the absence of power analyses made the utility of the Z statistic hard to interpret. Furthermore, the performance of the method was difficult to assess without positive controls or comparisons to other existing techniques. Concerns are described in the comments that follow.

## Major Concerns:

1. A proposed framework for the test(s) and statistical inference of the quantities of interest. The authors lay out a statistic and some motivation, and then how it could be used to infer specific quantities of interest, but the presentation was hard to grasp as the presentation was relatively informal. We would strongly encourage the authors to organize and present a formalized approach - inference of the statistic, the test in question, the null hypothesis (and interpretation), then a proposed test, how significance of the test will be assessed. 

1a. For example, the values of Z described in various figures in the manuscript are not easily interpretable -- what does the magnitude of Z mean? 

1b. What is the proposed test, and the null hypothesis? It seems the authors would mean Z > 1 is the alternative; so does that imply that Z < 1 is the null? What is the actual test used to determine that this is the case? 

1c. In the latter half of the paper, the authors calculated bootstrapped confidence intervals which may be the manner in which they would like to determine significance, but the procedure for these calculations were not well described. Again, it seems important to organize all of this into a single place with sufficient methodological details that the approach can be evaluated clearly (and is clear to the reader). 

1d. The simulation results were particularly glaring in this regard. It was not clear as to how this was done (i.e., everything should be in methods, organized succinctly with reasoning, rather than in legends). It seems important that there needs to be some sort of confidence interval estimate of variance for Z to assess if Z is meaningfully different from 1, and how this could be shown.

2. A formal, statistical workup of the proposed test(s) and inference. After the authors lay out a specific inference procedure, it seems incumbent then to more deeply explore the statistical characteristics of this statistic and their test. Without this, it is difficult to assess what the calculated scores actually quantify and to understand how well the proposed method works in practice.

2a. Power analyses. Is the method able to accurately identify balancing selection? Without a deeper interrogation of the power of the statistic, it is difficult to determine the utility of the method. While it is claimed that the method is robust with respect to generating false positives, no assessment is made to what extent the method is powerful enough to determine true positives. Parameters that might influence power could include (but are not limited to): survey sample sizes; TMRCAs, demographic effects; recombination and mutation rate variation; when a balanced allele arises and selection parameters that relate to its equilibrium frequency, different balancing selection models, etc.

2b. Error rate evaluation. While the authors present simulation and show the trends of various Z statistics under various demographic two-population scenarios, this feels like only a start to assess how often under various circumstances one might observe Z > 1 without balancing selection. Right now, the manuscript does not formally assess the error rate of the proposed approach (because there is no proposed test formally, and how often type 1 errors are made). 

2c. Properties of inference of number of balanced alleles. In particular, a statistical workup under the alternative for inferring the number of balanced alleles was notably absent in this presentation. They present estimates (with confidence) on real data, but without any workup where the truth is known, it is impossible to know how accurate/biased these estimates are in practice (and the authors admit for various reasons this approach may have bias). Simulations that answer this question would have significantly strengthened the claims that this method produces accurate estimates for both Z and the number of balanced polymorphisms. How far off is the method's estimate from the "true" number of balanced polymorphisms? Does the procedure to generate confidence intervals capture the truth at the expected level (90% CIs capture the true value 90% of the time)? Does this depend on the level of recombination or local LD (as the author's claim is a limitation in the introduction)?

2d. No comparison to previous methods. While the proposed method is claimed to excel where current methods struggle (robust to shorter divergence times), this presentation makes no attempt to compare to any existing methods and compare them to the proposed statistic despite the authors being aware of many methods (they cite many in the introduction). An evaluation on simulated data comparing performance would have strengthened the argument that this method is truly an improvement on currently existing methods, and thus a novel contribution to the field. 

3. Novelty. It could be that there is methodological or biological novelty here. However, the current presentation makes what is, in fact, novel extremely challenging to ascertain. 

3a. Defining methodological novelty. Is the proposed method simply McDonald-Kreitman? If not, the authors should explain exactly how/why (if the MK test is motivating) and point out the precise differences or the intuitive way they got to the inference procedure they presented. Moreover, the estimate of the proportion of balanced sites could be novel; however, without a workup of its merit (read: statistical assessment of accuracy etc.), it is hard to know if/how meaningful this is.

3b. Defining biological novelty. There is quite a bit of literature now which reports statistics that have been applied to 'scan' the human genome for selection. What, if any, loci do the authors highlight that have not been reported? No new genes were considered significant after Bonferroni corrections. There were no substantial discussions of new genes that were implicated with this new method. This itself diminishes the potential interest of the current work. However, if there exists some angles of novelty, we encourage the authors to really bring that forward in presentation. E.g. Specific examples, regional plots, to guide the intuition about the signal that the author's method is sensitive to (and perhaps why their method reveals it whilst others do not). 

4. Details about how the method was ultimately applied were missing or hard to find. What values were used for set parameters such as the choice of Watterson's coefficient or the proportion of neutral non-synonymous mutations? How were the number of neutral mutations maintained in LD set? Furthermore, the details for the bootstrapping procedure were also unclear. In data analysis, were windows used to analyze the data, and if so, what was considered (and why; does this choice influence power?)?

## Minor Comments:

5. The structure of the paper made it difficult to read. The logic of these types of papers approximately follow a theory, simulation/statistical workup, application to real data presentation flow. This manuscript has points that seem to jump across a lot of these spaces, making it confusing to read through.

6. Figures are difficult to interpret. Figure 1 was hard to parse and the figure caption confusing. The merging of descriptions for different panels was difficult to follow and the sharing of figure legends was not intuitive. Furthermore, the lack of variance on the simulation results, as described previously, detracted from the plotted point estimates. Several of the other figures were also a bit challenging to understand.

7. Point of discussion, in the intro it is true that several methods are computationally intensive and do require simulations and/or knowledge of demography to maximize power. However, not all methods do that. I would suggest some caution about how the authors report this (L50 - 52) different investigators might differently disagree with how they characterize their methods! (which is why the author's point about estimating the number of sites could add value). 

Reviewer #4:

"A new test demonstrates that balancing selection maintains hundreds of non-synonymous polymorphisms in the human genome", by Soni, Vos and Eyre-Walker is an interesting manuscript that addresses an interesting question: how many polymorphisms are maintained by balancing selection? As the authors note, how much variation is maintained by natural selection is a fundamental question in population genetics and evolutionary biology. The authors have modified a classical method to address this question. This is an excellent attempt at quantifying the effects of balancing selection. The authors then apply their method and conclude that balancing selection maintains more than a thousand non-synonymous polymorphisms in humans. This means that thousands of genes are under balancing selection in humans and that balancing selection is fairly common. This result is surprising as the current view in the field is that balancing selection is quite rare. If accurate the conclusions would be not only surprising, but also interesting and very important. They would have important implications and ramifications in our view of the evolution of polymorphism. I am very positive about this manuscript. Yet the authors do not show that they have explored the behaviour of the test in a way that establishes their estimates are clearly realistic. With a bit of extra work they can discard alternative possible explanations for their observations and make their results and their arguments convincing.

Major points:

- Introduction: The method is very different from existing methods in multiple ways. Z has many advantages over other methods, as stated in the manuscript. But also important disadvantages, the most important being:

1. That it is not adequate to identify individual genes, and 

2. That it cannot identify selection at non-coding loci. 

This should be more clearly stated, preferably in the introduction. 

For 2. the authors should mention how much of an issue this is expected to be. Are most balanced polymorphisms non-synonymous, or regulatory? What did the best previous methods that are not limited to non-synonymous changes find in this regard? 

- L 147: I understand the motivation to divide polymorphisms in bins of frequency. This has the advantage that the age of the alleles in the shared and private categories is similar, which I think addresses many potential issues. If the authors agree that this is an advantage, they could state this more clearly. But for shared polymorphisms the frequency used is the mean frequency in the two populations. This potentially moves the polymorphisms across bins of frequency compared with where they are in the population used for the private polymorphisms, in a way that is difficult for me to predict and that may affect K. Averaging could move non-synonymous polymorphisms to a higher frequency bin than they have in the population where private polymorphisms are counted, for example if purifying selection is less efficient in the other population. That would inflate Z in ways that seem consistent with patterns in Figure 2. But rather than me arguing about expectations, it is easy to explore how averaging frequency affects Z when populations have different Ne and whether this is an issue.

- I believe that LD to a balanced polymorphism must increase global Z. Not in the simulations because exons are individually simulated. But in real data where a single balanced polymorphism can maintain multiple non-synonymous variants through linkage in different exons of the same gene. This could potentially explain the higher Z in regions of low RR. If this is the case, the observed Z could be substantially overestimated. One could discard this possibility by analysing only one exon per gene.

- Figure 1: I cannot see that "Z increases as a function of the time since the population split". There is no change in many lines, and a modest one in others. Such a modest effect is surprising because as the authors mention one would expect to observe such an effect. 

- Demography influences Z. This being the case, in my opinion more efforts are needed to show that demography cannot explain the results, but under two simple models and a human demography they do not observe Z>1. Two issues I see:

1. The best approach would be to ask which models could potentially inflate Z and test them. The authors will be better at this than me, but many population size changes over time such as expansions and bottlenecks seem to have that potential. Simulation results for those two, and other models that can potentially inflate Z, would convince the reader that results are not due to differences in demography among two populations.

2. Even if demography could not by itself generate Z>1, by inflating Z it would affect the estimates of the number of non-synonymous polymorphisms maintained by balancing selection. Is this considered in the estimates presented?

- L 238: The human populations analysed are presented in terms of continental groups, but there are multiple populations in each of these groups. Was one population chosen per group, or was the combination of all populations in each of these groups analysed? In the latter case combining differentiated populations affects allele frequencies and could influence Z. I presume that the authors have considered this but it would be good to address this point and add this information to the methods.

- Figure 2: Are there no values for Eur for the Africa/Europe comparison for 0.1-0.5 in Figure 2? 

- Differences among populations and among analyses using a different population for the private polymorphisms are difficult to understand. I think that at least potential explanations should be provided. The case of EAS/SAS is odd. Z is positive on EAS but very negative in SAS. Are these two mirror images of the same pattern? Such pattern seems present in all plots and not expected from balancing selection as far as I can see. Could it be due to population differences in Ne? Or other technical effects? This is an important point needs to be addressed. I don't think that the discussion in L. 525 is convincing because it does not address this issue but points to a different argument, that in the limited set of simulated models Z was never >1.

- The analysis per gene category is very interesting and potentially informative. Many highlighted categories seem to contain very long genes, which is worrying. It would be easy to test if these genes are longer than expected or use a method that controls for gene length, to ensure that technical artifacts are not influencing the result.

- The analysis per gene seems so underpowered to become almost uninformative. First, what is the behaviour of the test when polymorphisms are binned as in this analysis? Second, as there are no significant signatures genes are chosen by how many times they have Z>1, but as noted in the text these analyses are not independent. I don't see much value in the analysis of individual genes but if the authors decide to keep it should start with a strong statement about the limitations of the method for this application.

- Associated overdominance is an interesting possible mechanism, but it is unclear why it is singled out if the authors do not think it is likely. I can guess, but it is not clear from the text.

Minor points:

- There are some track changes still in the text.

- The paper is well written and generally clear, but not easily accessible to a general PLoS Biology audience. The authors could explain more clearly the effects of demography and selection on allele frequencies and the test for the non-expert.

- More methods details need to be provided, especially in the Results. For example, in L122 the reader should be able to know what distribution of s is used, and more details on the demographic model used. 

- I am a bit confused about the effect of linkage on the method. My understanding is that linked alleles are not counted in the non-synonymous polymorphisms claimed to be maintained by balancing selection in the genome. But there must be non-synonymous polymorphisms that are maintained due to linkage to the polymorphism under balancing selection. Can the authors estimate the total number?

- Figures. I advice that titles are added to the figure, or at the very least the figure panels are numbered.

- The population size in the simulations is low. How strong was s so the balanced alleles were not lost due to drift?

---

## [Decision Letter · Decision Letter 2]

18 Jan 2022

Dear Adam,

Thank you for submitting a revised version of your manuscript "A new test of balancing selection and its application to data from humans" for consideration as a Research Article at PLOS Biology. This revised version of your manuscript has been evaluated by the PLOS Biology editors, the Academic Editor and three of the original reviewers. The Academic Editor has assessed your responses to reviewer #4, who was unable to re-review, and finds them satisfactory.

IMPORTANT:

a) Please attend to the remaining concerns from the reviewers.

b) After discussion with the Academic Editor, we think that your article would be best considered as a Methods paper, given its focus. When you re-submit, please can you change the article type to "Methods and Resources"?

In light of the reviews (below), we are pleased to offer you the opportunity to address the remaining points from the reviewers in a revised version that we anticipate should not take you very long. We will then assess your revised manuscript and your response to the reviewers' comments and we may consult the reviewers again.

We expect to receive your revised manuscript within 1 month.

**IMPORTANT - SUBMITTING YOUR REVISION**

*Resubmission Checklist*

*Published Peer Review*

*PLOS Data Policy*

*Blot and Gel Data Policy*

Sincerely,

Roli

Roland Roberts

Senior Editor

PLOS Biology

rroberts@plos.org

REVIEWERS' COMMENTS:

Reviewer #1:

I am satisfied with the authors' revision. I like the idea behind the method, and although the exact estimates of variants under balancing selection have to be taken with caution, I think the most critical issues of the method are thoroughly explored or discussed by the authors.

Reviewer #2:

This is a revised version of a paper that I reviewed earlier on. The new version has been improved and the authors have satisfactorily taken my comments into account. The "statistical test" part could have been a bit longer and it would have been nice to see the test applied to another species than humans but this does not diminish the novelty and the potential interest of the approach.

Minor comments:

Line 124: Yates et al, 2020 is missing from the Reference list

Line 256: remove "is".

Reviewer #3:

The authors provide thoughtful responses to many of the previous comments and expand simulations in several places to provide additional clarity about the approach and its properties, in particular demographic simulations to describe the expected value of Z, and the (biased) properties of the inference of alpha (the proportion of sites expected to be subject to balancing selection). 

That said, some lingering concerns remain given the current presentation. I still think the most obviously important part of the paper (and most novel) is around the estimate of alpha, even though it carries with it substantial bias (underestimated). 

My concern focuses here on the utility and capabilities of Z as a gene-level test. 

In the response to my comments and in the manuscript, the authors state that the method for single gene analysis is underpowered. I point out that there is still not a formal quantification of this. 

In response to this comment in the previous review (comment #2a), the authors seem to agree that many factors would influence power, but that this would "take a huge amount of work" which would "yield little insight" even though they state (without evidence) that they believe there are certainly situations "where the method has reasonable power". It also remains unclear if this approach does better than any previously published method, which brings with it concerns (in general) about utility.

To try to be fair to the authors here (and trying to step past particularities of how unconvincing this response is in this case), I suggest two approaches that ought to be straight-forward and would assuage some of my concerns over this point. My hope is that these should not be so onerous on the authors that they could not be accomplished. These would also allow the authors to be fair about the potential of their method, and readers could evaluate what is shared and determine if it is useful for their purposes. (In response to Comment 2D, the authors seem to argue that their method is "an addition to the methods available… underpowered to say anything about individual genes" but main utility could be around direct estimates of the number of polymorphism subject to balancing selection).

(i). Extensive simulation work has been undertaken which is very helpful. However, it seems that these report the E[Z] along with the associated standard error of the mean, which is so small as to be negligible).

Since they have the variance (needed for the error calculation), the authors could - without much extra work - present the distribution of Z under some simple basic demographic null (e.g., those with -ve selection Fig 1b say, and/or a small subset of the most important from Supplementary figure S1-S13) and alternative +ve balanced selection (Fig 1a, say). Intuitively, the more that alternative Z distribution overlaps with a given distribution of Z under a given demographic/-ve selection scenario without selection indicates reduced power; less overlap suggests more power. Thus, some description of the variance of Z in those scenarios would give some intuition about power even if one did not formally make calculations.

(ii). Many alternative methods exist. While perhaps a full characterization of power across all methods is certainly overkill (especially if this version of the method is not well powered), choosing a *single* one - an approach they could choose that minimizes difficult of use while maximizing prior power - with appropriate power analyses and assessing performance in comparison to the author's statistic would both add some quantification to the "underpowered" assertion, while also serving as a natural way to emphasize the novelty and impressive achievement of this statistic, namely the alpha estimate.

## additional minor comments

- L122 - what does "density of balanced polymorphism" mean? 

- What is the expected stationary allele frequency for figure 1a for the balanced site(s) given the parameters? They should report (esp. since they partition by allele frequency later).

- I appreciate the workup of S1-S13 presented. Given what they report, I have to admit that I am somewhat surprised that there does not appear to be a single scenario where the E[Z] is not greater than 1. In my experience, just about every form of selection has some confounding demographic scenario whereby a method that attempts to infer selection must try to reconcile. I scratched my head about this for a little while and couldn't immediately think of something. That said, I would strongly encourage the authors to think hard and out-of-box about scenarios that might actually results in Z > 1 in expectation. 

E.g., I feel like maybe recent introgression from a (say) hominid ancestor (something that data is increasingly indicating may be a possibility) might be one of those cases? 

- L792-794; L934-940; L941-944: Check double reference in citations

- L256: typo, extra 'is'

---

## [Editor Report · Decision Letter 3]

29 Mar 2022

Dear Adam,

Thank you for submitting your revised Methods and Resources paper entitled "A new test of balancing selection and its application to data from humans" for publication in PLOS Biology. The Academic Editor and I have now assessed your responses and revisions. 

Based on this assessment, we will probably accept this manuscript for publication, provided you satisfactorily address the following data and other policy-related requests.

IMPORTANT:

a) I wonder if we could "have our cake and eat it" in the Title, i.e. flag both the method and the intriguing findings? I suggest the following: "A new test applied to human data reveals hundreds of non-synonymous polymorphisms subject to balancing selection." If you're happy with this, then please change it; if not, do suggest something that achieves the same end but that you're comfortable with!

b) My understanding is that the vast majority of your main and supplementary Figs (Figs 1, S1-S35) can be generated using the code/data that have been deposited in Github (https://github.com/vivaksoni/test_for_balancing_selection). If so, please include an explicit statement to this effect (with the Github URL) in each main and supp Fig legend.

c) ...but for Figs 2, S36, S37, S38 we probably need the data that underlie the Fig panels to be provided as separate supplementary data files (or sheets within a single file, e.g. S3_data.xslx). You should provide these, and indicate in each relevant Fig legend "The data underlying this Figure can be found in S3 Data." Let me know if you have a more appropriate solution to our data policy requirements.

We expect to receive your revised manuscript within two weeks. 

*Published Peer Review History*

*Press*

Best wishes,

Roli

Senior Editor,

rroberts@plos.org,

PLOS Biology

DATA POLICY:

Regardless of the method selected, please ensure that you provide the individual numerical values that underlie the summary data displayed in the following figure panels as they are essential for readers to assess your analysis and to reproduce it: Figs 2, S36, S37, S38. NOTE: the numerical data provided should include all replicates AND the way in which the plotted mean and errors were derived (it should not present only the mean/average values).

DATA NOT SHOWN?

---

## [Editor Report · Decision Letter 4]

25 Apr 2022

Dear Adam,

On behalf of my colleagues and the Academic Editor, Nick Barton, I'm pleased to say that we can in principle accept your Methods and Resources paper "A new test suggests hundreds of amino acid polymorphisms in humans are subject to balancing selection" (yes, that title works for us!) for publication in PLOS Biology, provided you address any remaining formatting and reporting issues. These will be detailed in an email that will follow this letter and that you will usually receive within 2-3 business days, during which time no action is required from you. Please note that we will not be able to formally accept your manuscript and schedule it for publication until you have completed any requested changes.

Sincerely,

Roli 

Roland G Roberts, PhD 

Senior Editor 

PLOS Biology

rroberts@plos.org